# Effect of Food Intake on Exhaled Volatile Organic Compounds Profile Analyzed by an Electronic Nose

**DOI:** 10.3390/molecules28155755

**Published:** 2023-07-30

**Authors:** Silvano Dragonieri, Vitaliano Nicola Quaranta, Andrea Portacci, Madiha Ahroud, Marcin Di Marco, Teresa Ranieri, Giovanna Elisiana Carpagnano

**Affiliations:** Department of Respiratory Diseases, University of Bari “Aldo Moro”, 70121 Bari, Italy; vitalianonicola.40@gmail.com (V.N.Q.); a.portacci01@gmail.com (A.P.); madiha.ahroud@gmail.com (M.A.); marcindimarco@gmail.com (M.D.M.); teresa.ranieri@uniba.it (T.R.); elisiana.carpagnano@uniba.it (G.E.C.)

**Keywords:** volatile organic compounds, electronic nose, e-nose, breathomics, breath analysis

## Abstract

Exhaled breath analysis using an e-nose is a groundbreaking tool for exhaled volatile organic compound (VOC) analysis, which has already shown its applicability in several respiratory and systemic diseases. It is still unclear whether food intake can be considered a confounder when analyzing the VOC-profile. We aimed to assess whether an e-nose can discriminate exhaled breath before and after predefined food intake at different time periods. We enrolled 28 healthy non-smoking adults and collected their exhaled breath as follows: (a) before food intake, (b) within 5 min after food consumption, (c) within 1 h after eating, and (d) within 2 h after eating. Exhaled breath was collected by a formerly validated method and analyzed by an e-nose (Cyranose 320). By principal component analysis, significant variations in the exhaled VOC-profile were shown for principal component 1 (capturing 63.4% of total variance) when comparing baseline vs. 5 min and vs. 1 h after food intake (both *p* < 0.05). No significance was shown in the comparison between baseline and 2 h after food intake. Therefore, the exhaled VOC-profile seems to be influenced by very recent food intake. Interestingly, two hours might be sufficient to avoid food induced alterations of exhaled VOC-spectrum when sampling for research protocols.

## 1. Introduction

Breathomics, also known as breath analysis or breath metabolomics, is a rapidly emerging field in the realm of medical research and diagnostics. It involves the analysis of volatile organic compounds (VOCs) present in a person’s breath to obtain insights into their physiological and pathological conditions. By analyzing the unique composition of VOCs, breathomics aims to provide non-invasive and real-time information about an individual’s health status [1].

The concept of breath analysis dates back centuries, but recent advancements in technology and analytical techniques have allowed for more precise and comprehensive analysis of breath samples. Breathomics relies on highly sensitive instruments such as gas chromatography-mass spectrometry (GC-MS), proton transfer reaction-mass spectrometry (PTR-MS), and electronic nose (e-noses, Figure 1) devices to detect and identify a wide range of VOCs [2]. The human breath contains a complex mixture of VOCs, which are metabolic byproducts of various physiological processes occurring within the body [3]. These compounds can originate from multiple sources, including the respiratory system, gastrointestinal tract, liver metabolism, and systemic circulation. Changes in the concentration and pattern of VOCs can reflect alterations in metabolic pathways, oxidative stress, inflammation, and the presence of certain diseases or conditions [3].

Breathomics has the potential to revolutionize healthcare by offering several advantages over traditional diagnostic methods. It is non-invasive, painless, and easily repeatable, making it suitable for regular monitoring of patients and screening of large populations. Additionally, breath samples can be collected at any time and in various settings, providing flexibility and convenience for both patients and healthcare professionals.

The applications of breathomics are wide-ranging. It holds promise in the early detection and monitoring of diseases such as lung cancer, asthma, chronic obstructive pulmonary disease (COPD), gastrointestinal disorders, metabolic disorders, and infectious diseases [1,4]. It can also be utilized in personalized medicine, drug development, and therapeutic monitoring. Despite the exciting prospects of breathomics, there are still challenges to overcome. Standardization of sampling techniques, data analysis, and the identification of specific VOC biomarkers for different diseases are ongoing areas of research. Among several issues, it is still unclear whether food intake can be considered a confounder when analyzing the VOC-profile. Based on the above, we aimed to assess whether an e-nose can discriminate exhaled breath before and after predefined food intake at different time periods.

## 2. Results

Baseline characteristics of the study population are shown in Table 1. No sex imbalances were reported, and lung function parameters and body mass indexes (BMI) were also normal. By principal component analysis (PCA), significant variations in the exhaled VOC profile were shown for principal component 1 (capturing 63.4% of total variance) when comparing baseline vs. 5 min and vs. 1 h after food intake (*p* < 0.05 for both analyses, see Table 2 and Figure 2). No significance was shown in the comparison between baseline and 2 h after food intake (*p* = ns, see Table 2 and Figure 2). In the PCA plot, breathprints of subjects at baseline differed from those after 10 min of food intake (Figure 3). Subsequent canonical discriminant analysis revealed a Cross Validated Accuracy (CVA) of 65%, *p* < 0.05. The AUC of the ROC curve for the discrimination between baseline and 10 min after food intake was 0.632. Similarly, breathprints at baseline were separated from those after 1 h of food intake (Figure 4). Subsequent canonical discriminant analysis revealed a CVA of 65%, *p* < 0.05. The AUC of the ROC curve for the discrimination between baseline and 1 h after food intake was 0.703. When comparing exhaled VOC-profile of baseline vs. 2 h after food intake, no separations were shown, with a CVA of 47.5%, *p* = ns (Figure 5).

## 3. Discussion

Our study reveals that the analysis of exhaled volatile organic compound (VOC) profiles using an electronic nose (e-nose) is influenced by recent food consumption. Interestingly, two hours might be sufficient to avoid food-induced alterations of exhaled VOC-spectrum when sampling for research protocols. In recent years, e-nose-based analysis of exhaled breath has gained popularity for screening and diagnostic purposes in various medical conditions [1,4]. Breath analysis offers advantages over other biological samples like blood, feces, and urine, as it is non-invasive, easily obtainable, and enables real-time monitoring. Nonetheless, the field continues to evolve rapidly, and ongoing advancements in technology and data analysis are expected to unlock the full potential of breathomics in the near future, leading to improved diagnostics and personalized healthcare.

The novelty of our study lies in assessing the VOC composition of exhaled breath using e-nose technology in relation to short-term food exposure. Previous investigations about food effect on exhaled VOCs were performed by using GC-MS. In particular, Raninen et al. showed that a high-fiber diet reduced the fasting level of several VOCS, especially of exhaled 2-methylbutyric acid and the postprandial response of 1-propanol [5]. Five years later, the same group extended their observations by detecting 260 VOCs from exhaled breath samples and evidencing derivatives of benzoic acid and phenolic compounds, as well as some furanones only after whole grain diets [6]. Moreover, Wuthrich et al. analyzed the chemical composition of the human postprandial breath metabolome before and after the ingestion of a standardized nutritional shake, demonstrating modifications of fatty acids, amino acids, and unsaturated hydrocarbons, as well as molecules related to gut microbiome activity and the citric acid cycle amino acid derivatives [7]. Finally, Jaksic and colleagues developed of a novel bioanalytical method for the assessment of food impact on selected exhaled VOCs using a fast and portable screening VOC prototype sensor based on membrane inlet mass spectrometry, showing large modifications in acetone, isoprene, and n-pentane levels 120 min after the meal, as well as ethanol levels, after 60 min [8]. Our findings align with these previous observations, evidencing that the whole exhaled VOC-spectrum seems to be influenced by very recent food intake.

We focused on a well-characterized group of healthy subjects who were never smokers, with no known diseases. Notably, smoking was strictly forbidden due to its known impact on exhaled VOCs spectrum [9]. The methods used for e-nose analysis were standardized [10], and all participants consumed exactly the same food, which was mixed carbohydrate, fat, and protein content. However, there are a few limitations to consider. Firstly, the sample size of our study is relatively small. Although we believe that our population of 28 individuals justifies further investigation based on previous observations and our sample size estimation, larger cohorts and validation groups should be included in future studies.

Secondly, based on previous investigations [8], we chose arbitrary sampling periods after eating, and we lack data on longer periods post-meal, which might have provided additional relevant information.

Thirdly, while e-nose analysis is non-invasive, user-friendly, and provides quick results, it does not allow for the identification and quantification of individual VOCs. Therefore, further research should integrate chemical analytical techniques such as GC-MS to identify specific VOCs that can differentiate between groups.

How can we explain our findings? Previously, researchers have examined the changes in certain exhaled breath volatile organic compounds (VOCs) following a meal or the ingestion of glucose [11,12,13]. These VOCs include acetone, ammonia, isoprene, methanol, ethanol, propanol (mainly isopropanol), methyl nitrate, xylene, ethyl benzene, propionic acid, and butanoic acid [11,12,13]. However, previous studies have shown conflicting results regarding VOC changes, which might be attributed to variations in individual metabolic responses and the body’s microbiota [14], as well as for remnants of ingested food and bacterial fermentation occurring in the mouth or throat [13]. Undoubtedly, it is crucial to take into account the duration after consuming food as a significant variable that can impact the composition of volatile organic compounds (VOCs) in breath gas analyses. Therefore, when conducting VOC analyses, it is essential to follow standardized sampling protocols that consider both the timing of food consumption and the specific properties of the food consumed across different people and races.

In conclusion, our data indicate that very recent food consumption can act as a confounding factor influencing e-nose measurements. Therefore, we recommend that patients abstain from eating for not less than two hours before undergoing exhaled breath analysis using e-nose technology. It is necessary to conduct future research using larger groups of participants, varying their diet and lifestyle habits. This research should consider including individuals with metabolic syndrome, diabetes, and cardiovascular problems. By doing so, we anticipate obtaining a more comprehensive understanding of the effects of food on breath VOCs. Additionally, it will shed light on potential connections between physical activity, diet types, and other factors with specific breath VOCs. Furthermore, it is important for different e-nose technologies to confirm and build upon our findings, as well as to investigate other factors that could potentially influence the results.

## 4. Materials and Methods

### 4.1. Patients

A total number of 28 healthy volunteers (14 males, 14 females) participated to our investigation. All individuals were free from clinical history of chest symptoms and/or systemic diseases, and none of the participants were assuming pharmacological therapy. Age range was 28–53. All subjects had lung function within normal ranges. Participants with history of upper or lower respiratory tract infections in the four weeks before investigations were eliminated from the study. Exhaled breath was collected from all individuals as follows: (a) before food intake (T0), (b) within 5 min after food consumption (T1), (c) within 1 h after eating (T2), and (d) within 2 h after eating (T3). All subjects were recruited from hospital staff and participation was on a voluntary basis. The present study was previously approved by the ethics committee of Bari Policlinico (protocol number 46403/15), and all participants signed an informed consent form before participating in the study.

### 4.2. Study Design

We performed a longitudinal study. Participants were called for two separate visits to complete all measurements. During their first visit, all subjects were screened for inclusion/exclusion criteria and, after being included in the study, a flow-volume spirometry was performed (MasterscreenPneumo, Jaeger, Würzburg, Germany).

During their second visit, exhaled breath was collected when fasting for at least 12 h (T0) and after eating a standardized meal composed by two slices of wholewheat bread (ingredients: whole wheat flour, water, sugar, wheat gluten, soybean oil, yeast, molasses, oat fiber, salt, monoglycerides, calcium propionate, sorbic acid, sodium stearoyl lactylate, soy lecithin, whey, citric acid and butter. Total weight 19.7 g, energy 54 kcal, fats 1.1 g, carbohydrates 8.5 g, fibers 1.4 g, proteins 1.8 g), two slices of baked ham (ingredients: pork meat, salt, natural flavorings, dextrose, sodium ascorbate and sodium nitrite. Total weight 40 g, energy 86 kcal, fats 6.1 g, carbohydrates 0.4 g, fibers 0 g, proteins 8.2 g), two slices of Italian cheese (ingredients: cow milk, salt, rennet. Total weight 40 g, energy 124 kcal, fats 9.2 g, carbohydrates 0.4 g, fibers 0 g, proteins 9.3 g) and a thin layer of mayonnaise (ingredients: sunflower oil, water, white wine vinegar, pasteurized egg yolk, salt, modified starch, xanthan gum; concentrated lemon juice, flavorings Total weight 5 g, energy 12.7 kcal, fats 1.1 g, carbohydrates 0.3 g, fibers 0 g, proteins 0.03 g) at T1, T2, and T3 as described above and analyzed by the e-nose straightaway. All participants were allowed to drink only still water during the day of testing.

Exhaled breath was collected as previously described [10] by wearing a noseclip during the whole procedure; firstly, a 5 min wash-in period with a 3-way non-rebreathing valve coupled to an inspiratory VOC filter (A2; North Safety, Middelburg, The Netherlands) to minimalize the impact of environmental VOCs. Successively, subjects exhaled a vital capacity into a Tedlar bag which was immediately sampled by the e-nose.

### 4.3. Electronic Nose

We utilized a commercially available electronic nose (Cyranose 320, Sensigent, Irwindale, CA, USA) for our study. This device consists of a collection of 32 organic polymer sensors arranged in a nano-composite array. When exposed to combinations of volatile organic compounds (VOCs), these polymers undergo swelling, leading to changes in their electrical resistance. The device records the raw data (expressed by dR = (R − Ro)/Ro where R is the response of the system to the sample gas, and Ro is the baseline reading, the reference gas being the ambient room air) in each of the 32 individual sensors and stores it in an onboard database. This data forms a unique “breathprint” that represents the VOC spectrum and can be further analyzed using pattern-recognition algorithms (see Figure 1). To ensure accurate measurements, we followed the instruction manual’s recommended operating parameters. The baseline purge was set at 30 s with a low pump speed. The sampling time was 60 s with a medium pump speed. A high pump speed was used for a 200-s purging time. The total run time for each sample was 300 s, and the device was operated at a temperature of 42 °C. Between samples, a post-run purge of 5 min was performed. Additionally, before taking the first sample of the day, the sensors were stabilized by exposing them to room air for 5 min followed by a “blank measurement”. During the analysis of exhaled samples, the relative humidity was approximately 55%.

### 4.4. Statistical Analysis

All data analyses were performed using SPSS for Windows 26.0 (SPSS, Chicago, IL, USA). The Kolmogorov–Smirnov demonstrated a non-parametric distribution of data. Categorical values were analyzed using the chi-square test or Fisher’s exact test as appropriate and were reported as n (%). Median (interquartile range [IQR]) was used to express continuous variables with nonparametric distribution. Data were reduced by principal component analysis. All four Principal Components (PCs) were compared at the aforementioned different times (T0, T1, T2 and T3) by Kruskal–Wallis test. Where it showed the significance, individual groups were compared two to two by the Mann–Whitney U test and Shapiro–Wilk tests. Following that, successive linear canonical discriminant analysis (CDA) was carried out to obtain the cross-validated accuracy percentage (CVA%). This percentage serves as an estimate of how effectively a predictive model will perform in real-world scenarios. Furthermore, a receiver operating characteristic curve (ROC-curve) was constructed using predicted probabilities in order to calculate the area under the curve (AUC). To ensure a manageable standard error of 10%, we determined the appropriate sample size for our study. Taking into account an estimated accuracy rate of 80%, the current sample size per subgroup was deemed sufficient. We established a threshold for statistical significance with a *p*-value of <0.05.

## Figures and Tables

**Figure 1 molecules-28-05755-f001:**
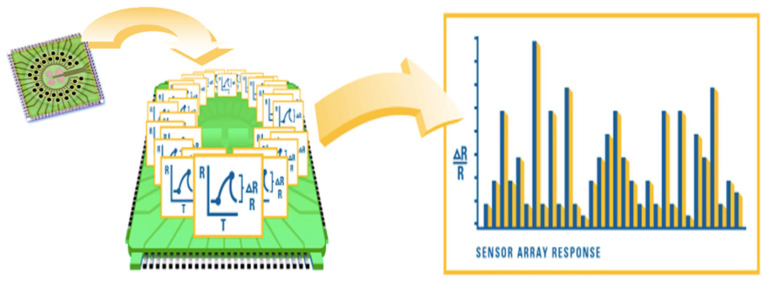
Working principle of Cyranose 320. It is based on a nano-composite array of 32 organic polymer sensors. If exposed to VOC combinations, the polymers swell, thereby modifying their electrical resistance. Raw data are registered as the increase in resistance of any single sensors and the combination of all signals results in a “breathprint”.

**Figure 2 molecules-28-05755-f002:**
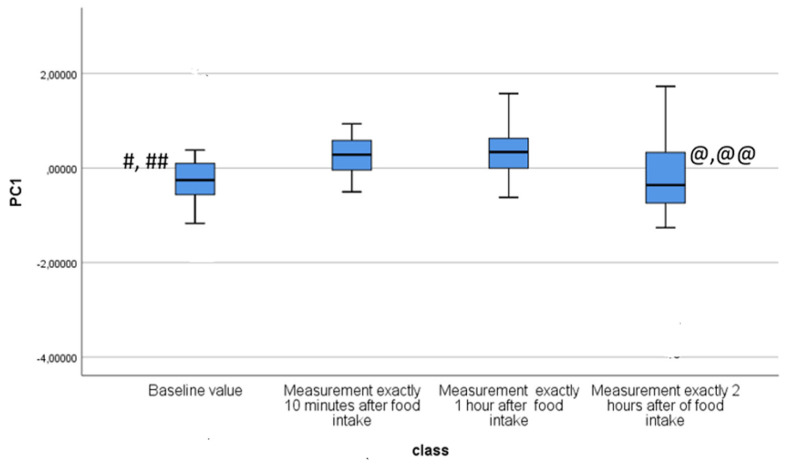
Box plot showing comparisons of PC1 among different times. *p* < 0.050 for the following: #: Baseline vs. 10 min after food intake; ##: Baseline vs. 1 h after food intake; @ 10 min after food intake vs. 2 h after food intake; @@: 1 h after food intake vs. 2 h after food intake.

**Figure 3 molecules-28-05755-f003:**
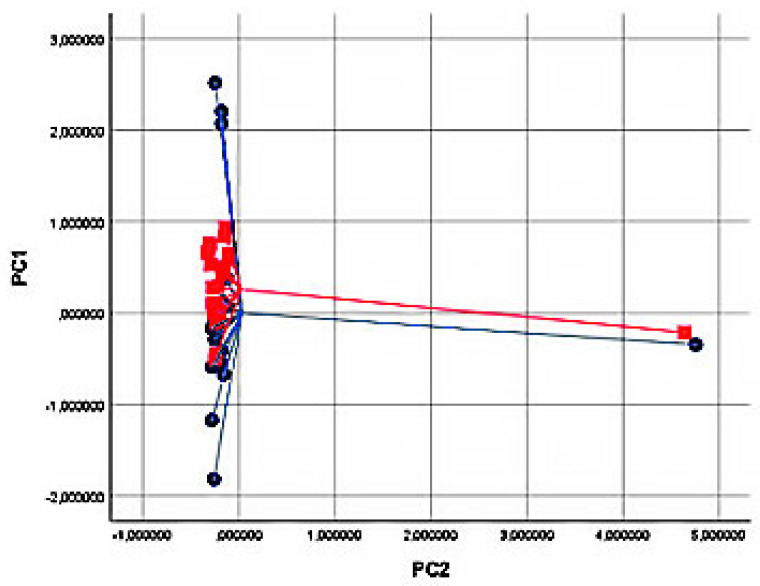
Two-dimensional Principal Component analysis plot showing the discrimination between breathprints of baseline (T0, blue bullets) and 10 min after food intake (T1, red squares). Subsequent Discriminant analysis showed a Cross Validated Accuracy of 65%.

**Figure 4 molecules-28-05755-f004:**
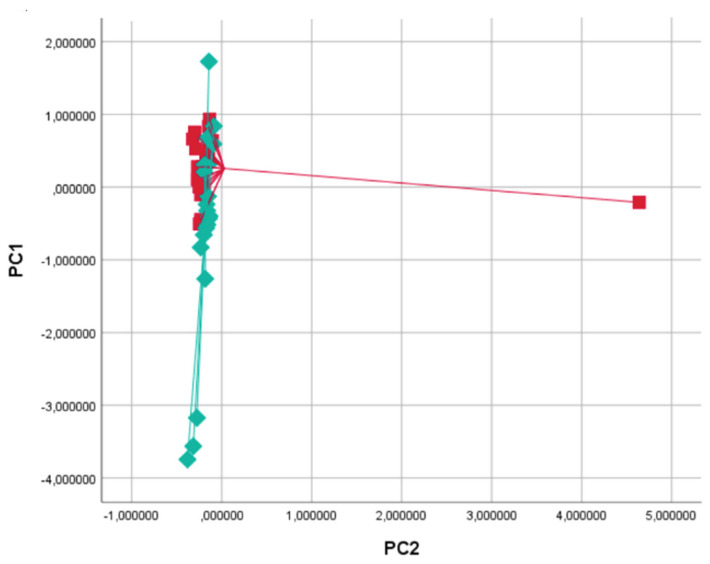
Two-dimensional Principal Component analysis plot showing the discrimination between breathprints of baseline (T0, red squares) and 1 h after food intake (T2, green diamonds). Subsequent Discriminant analysis showed a Cross Validated Accuracy of 65%.

**Figure 5 molecules-28-05755-f005:**
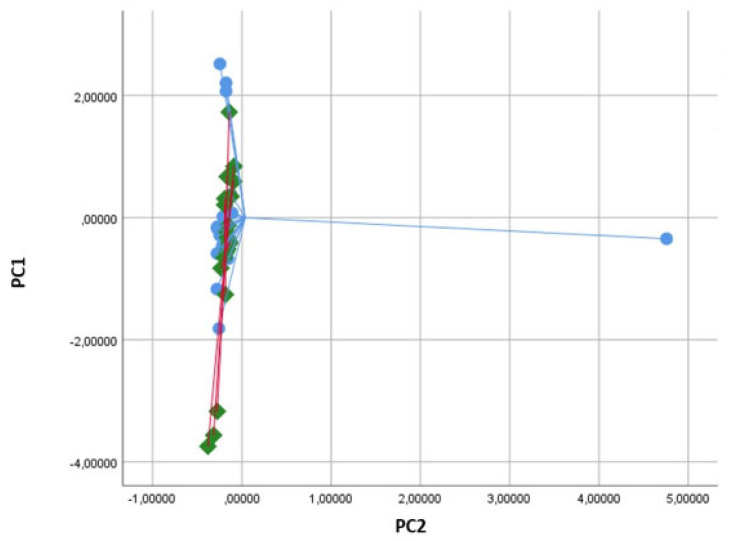
Two-dimensional Principal Component analysis plot showing the non-discrimination between breathprints of baseline (T0, blue bullets) and 2 h after food intake (T3, green diamonds). Subsequent Discriminant analysis showed a Cross Validated Accuracy of 45%.

**Table 1 molecules-28-05755-t001:** Clinical characteristics of the studied population.

Parameter	Value
Subjects (n.)	28
M/F (n.)	14/14
Age (y.)	35.2 ± 10.8
FEV1%pred.	100.8 ± 10.5
BMI (kg/m^2^)	26.1 ± 2.9
smokers (n.)	0
Comorbidities (n.)	0

FEV1 = forced expiratory volume in the 1st second.

**Table 2 molecules-28-05755-t002:** Principal Component Analysis on the dataset. PC1 collected 63.4% of the total variance whereas PC2 collected 29.2% of the variance.

	Baseline (T0)	10 min after Food Intake (T1)	1 h after Food Intake (T2)	2 h after Food Intake (T3)	*p*
PC1 median (IQ 25–75) *^,#,@,§^	−0.256 (−0.570–0.106)	0.282 (−0.068 0.611)	0.337 (−0.031 0.644)	−0.361 (−0.786 0.336)	0.008
PC2 median (IQ 25–75)	−0.216 (−0.264–0.147)	−0.216 (−0.264–0.147)	−0.172 (−0.218–0.117)	−0.166 (−0.197–0.142)	0.081
PC3 median (IQ 25–75)	−0.889 (−0.383 0.198)	0.138 (−0.032 0.407)	0.128 (−0.069 0.306)	−0.173 (−1.130 1.374)	0.185
PC4 median (IQ 25–75)	−0.060 (−1.703 0.125)	0.080 (−0.251 0.182)	0.121 (−0.002 0.236)	−0.577 (−0.432 0.548)	0.182

IQ 25–75: inter quartile 25–75. * = PC1: comparison between T0 and T1: *p* = 0.013; ^#^ = PC1: comparison between T0 and T2: *p* = 0.015; ^@^ = PC1: comparison between T1 and T3: *p* = 0.018; ^§^ = PC1: comparison between T2 and T3: *p* = 0.015.

## Data Availability

Anonymized raw data is available at request.

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
