# Peer review of "Effect of Food Intake on Exhaled Volatile Organic Compounds Profile Analyzed by an Electronic Nose"

_molecules, 2023, doi:10.3390/molecules28155755_

Round 1

Reviewer 1 Report

This paper studied the influence of food intake on the detection of VOCs in exhaled breath. The author mainly studied the influence of food intake time on the analysis results through electronic nose, and obtained some useful conclusions. However, the work is relatively simple, working more like a science report or an experiment report. The authors need to increase experimental design and analysis to improve the reliability of results and research depth. Therefore, I do not recommend accepting this work for publication.

Author Response

R- Dear reviewer, thanks for your time spent in evaluating our manuscript. We do respect your opinion and in the revised version we have now improved the data analysis section by adding PCA plots and discriminant analysis results, as well as added new references.  

Reviewer 2 Report

General Comments:

 The purpose of the work is clear and simple: To assess how long the breath exhalation of VOCs is influenced by food intake. The methodology is consistent and the manuscript is well written. However the reader misses the presentation of some intermediate results and plots in order to understand the meaning of Table 2 and Figure 2, and be convinced that two hours are enough to “clean” the breath exhalation after food intake. Results should be further described by answering / including:

 - What was the sensor response attribute extracted for analysis? (delta R/Ro, I suppose).

- How many attributes were used? (32, I suppose, because there are 32 sensors in the sensor array).

- Include the PCA plot (PC1 x PC2) so the reader can “see” the separation among the samples (T1 far from T0) and (T3 close to T0).

Specific Comments:

 - The keyword "food" is too generic. I suggest replacing it with "breath analysis" and "breathomics".

Author Response

The purpose of the work is clear and simple: To assess how long the breath exhalation of VOCs is influenced by food intake. The methodology is consistent and the manuscript is well written. However the reader misses the presentation of some intermediate results and plots in order to understand the meaning of Table 2 and Figure 2, and be convinced that two hours are enough to “clean” the breath exhalation after food intake. Results should be further described by answering / including:

 - What was the sensor response attribute extracted for analysis? (delta R/Ro, I suppose).

R- Thanks for your comments. You are right, it’s delta R/Ro. We have now specified it in the manuscript.

- How many attributes were used? (32, I suppose, because there are 32 sensors in the sensor array).

R- Thanks for the question. Yes we used 32 attributes, the exact number of cyranose sensors. We have now specified it in the paper.

- Include the PCA plot (PC1 x PC2) so the reader can “see” the separation among the samples (T1 far from T0) and (T3 close to T0).

R- Many thanks for this comment. We have now added PCA plots and discriminant analysis results. Please see data analysis, results section and new figure 3, 4 and 5.

Specific Comments:

 - The keyword "food" is too generic. I suggest replacing it with "breath analysis" and "breathomics".

R- Done it. Thanks.

Reviewer 3 Report

The manuscript looks like a report of experimental results only. No detailed data about intaken food.  Low-quality figures and a very short list of references. The conclusions are rather symbolic. I recommend rejecting it.

Author Response

R- Dear reviewer, thanks for your time spent in evaluating our manuscript. We do respect your opinion and in the revised version we have now included more details about food, improved the data analysis section with PCA plots and discriminant analysis results. Moreover, we have improved our figures with better quality, as well as new references.

Round 2

Reviewer 1 Report

1. Don't include a period in the title;

2. Some keywords in the abstract should be deleted, such as Background., Aim. Etc.;

3. The charts all need to be redone in accordance with the standard format.

Author Response

Dear reviewer, thanks for changing your opinion after evaluating our revised version. Moreover, in R2 we have now added detailed nutritional infos about the standardized meal (please see methods section). 

  1. Don't include a period in the title.      R- Removed. Thanks. 

2. Some keywords in the abstract should be deleted, such as Background., Aim. Etc.; R- Deleted. Thanks.

3. The charts all need to be redone in accordance with the standard format. R- Many thanks for your observation. This will be done during proof correction stage, with the help of editorial office, so that we are sure about the right style to apply.